# Motor Adaptation Deficits in Children with Developmental Coordination Disorder and/or Reading Disorder

**DOI:** 10.3390/children11040491

**Published:** 2024-04-19

**Authors:** Jérémy Danna, Margaux Lê, Jessica Tallet, Jean-Michel Albaret, Yves Chaix, Stéphanie Ducrot, Marianne Jover

**Affiliations:** 1CLLE, Université de Toulouse, CNRS, 31058 Toulouse, France; 2Aix-Marseille University, PsyCLE, 13284 Aix-en-Provence, France; margaux.le@univ-amu.fr (M.L.); marianne.jover@univ-amu.fr (M.J.); 3Aix-Marseille University, CNRS, CRPN, 13015 Marseille, France; 4ToNIC, Université de Toulouse, Inserm, UT3, 31300 Toulouse, France; jessica.tallet@inserm.fr (J.T.); chaix.y@chu-toulouse.fr (Y.C.); 5Pediatric Neurology Department, Children’s Hospital, Toulouse University Hospital, 31300 Toulouse, France; 6Aix-Marseille University, CNRS, LPL, 13100 Aix-en-Provence, France; stephanie.ducrot@univ-amu.fr

**Keywords:** procedural perceptual-motor learning, handwriting, neurodevelopmental disorders, comorbidity

## Abstract

Procedural learning has been mainly tested through motor sequence learning tasks in children with neurodevelopmental disorders, especially with isolated Developmental Coordination Disorder (DCD) and Reading Disorder (RD). Studies on motor adaptation are scarcer and more controversial. This study aimed to compare the performance of children with isolated and associated DCD and RD in a graphomotor adaptation task. In total, 23 children with RD, 16 children with DCD, 19 children with DCD-RD, and 21 typically developing (TD) children wrote trigrams both in the conventional (from left to right) and opposite (from right to left) writing directions. The results show that movement speed and accuracy were more impacted by the adaptation condition (opposite writing direction) in children with neurodevelopmental disorders than TD children. Our results also reveal that children with RD have less difficulty adapting their movement than children with DCD. Children with DCD-RD had the most difficulty, and analysis of their performance suggests a cumulative effect of the two neurodevelopmental disorders in motor adaptation.

## 1. Introduction

In a seminal article, Doyon and Ungerleider [1] made a distinction between two categories of procedural perceptual–motor learning: motor sequence learning and motor adaptation. Motor sequence learning refers to the acquisition of new motor sequences, or to the progressive acquisition of movements towards a well-executed behavior. It is experimentally explored through tasks wherein subjects are required to produce a sequence of movements, to discover a particular sequence by stimulus-response associations, or by repeating movements. Motor adaptation concerns the capacity to compensate for environmental constraints or changes, or to modify an internal motor representation. During the learning process, motor skills become progressively performed effortlessly through repeated practice, which produces representational changes in neural networks. According to this model [1], both categories of procedural learning would recruit similar cerebral structures in the early phase (motor, prefrontal and parietal cortical areas, striatum, and cerebellum). Once learned, the skill representation would be distributed in a network of structures that involves either the cortico-striatal system in the case of motor sequence learning, or the cortico-cerebellar system in the case of motor adaptation [2,3]. Motor learning requires both motor adaptation and sequence learning. However, depending on the skill and the learning phase, one system could be more recruited than the other.

Nicolson and Fawcett [4,5] proposed that procedural learning deficit could constitute the core underlying dysfunction in neurodevelopmental disorders and explain the frequent comorbidity between Developmental Coordination Disorder (DCD) and dyslexia. Considering dyslexia as a reading disorder, and for the sake of clarity, we have opted throughout the manuscript for the term “Reading Disorder” (RD), which is a better counterpart to DCD and consistent with the terms used in the DSM-5. Whilst children with DCD have difficulty performing age-appropriate perceptual–motor skills, children with RD present difficulties in word recognition that hinder their scholarship, both in the absence of diagnosable neurological disorders or adverse circumstances [6]. Despite the heterotopic nature of this comorbidity, Nicolson and Fawcett [5] assumed that a deficit in the procedural memory system could subtend altogether the impaired cognitive, motor, and linguistic abilities. Several meta-analyses published in the past ten years are in line with this view, and have concluded that procedural learning was frequently impaired in neurodevelopmental disorders [7,8,9,10,11]. In DCD, Biotteau et al. [8] concluded on the presence of procedural learning difficulties, despite many discrepancies between the studies—probably related to the heterogeneity of the disorder and the difference between the learning tasks. In dyslexia, Lum et al. [11] concluded on a global reduction in procedural learning ability.

Nevertheless, most of these conclusions rely on studies that examine procedural learning focusing on motor sequence learning. Several tasks have been investigated, such as serial reaction time task (SRTT) [11,12,13], the learning of new coordination between hands in a synchronization paradigm (finger tapping tasks [14,15,16]), or the learning of complex sequential movements in more ecological tasks [17,18,19,20]. For instance, Huau et al. [17] compared children with and without DCD using a task in which children had to learn and produce a new letter. They observed a lower quality and a higher variability in children with DCD as compared to typically developing (TD) children, and argued in favor of a deficit in motor pattern stabilization in children with DCD.

Studies focusing on motor adaptation in neurodevelopmental disorders are scarcer and reveal more discrepancies. Lejeune et al. [21] tested children with and without DCD using an inverted mouse task. They showed similar rates of learning, consolidation, and transfer in DCD and TD children, even if the DCD children’s performance remained lower and their difficulties did not diminish with practice. Kagerer et al. [22] exposed children to a rotation of their visual feedback while they performed a drawing task. They observed that DCD children were less affected than their typically developing (TD) peers, but also demonstrated fewer aftereffects. Cantin et al. [23] used a prism adaptation task in children with DCD, but failed to observe a significant impairment of adaptation in their small group (nine children). Using the same paradigm, Gómez-Moya et al. [24] observed less adaptation and smaller aftereffects in children with DCD when displacing objects with a reversing prism. Finally, concerning RD, in a study investigating both motor sequence learning (SRTT) and motor adaptation (mirror drawing), deficits were observed in children with RD compared to children without RD on both tasks [25].

Distinguishing between the two categories of procedural learning and their alteration in various neurodevelopmental disorders could contribute to a better understanding of the underlying mechanisms involved. For instance, Nicolson and Fawcett [4,5] assumed that a cortico-cerebellar dysfunction could be specific to RD, and a cortico-striatal dysfunction specific to DCD. Following Doyon and Ungerleider’s model [1], the cortico-striatal system should be recruited in the case of motor sequence learning and the cortico-cerebellar system in the case of motor adaptation. However, despite some discrepancies, the results presented above demonstrate that both categories of procedural learning were impacted in RD and DCD. Moreover, the authors often failed to check whether the children in their sample had any DCD associated with RD, and vice versa [25]. Studying RD or DCD without controlling for the presence of the other disorder strongly limits the conclusion.

Although comorbidity in neurodevelopmental disorders is common and has received increasing interest since the beginning of the 2000s [26,27,28,29], procedural learning has been rarely explored in children with associated DCD and RD. Even if, despite their respective specificities, children with isolated RD or DCD share common core difficulties, children with the comorbid condition should present with a cumulative impact on their cognitive abilities. In studies comparing children with DCD, RD, and DCD-RD, the results are not so clear cut. The performances of children with DCD-RD sometimes mirror those of children with RD [30], or sometimes those of children with DCD [31]. They may also present a simple addition of RD and DCD difficulties without any increase in severity [32,33,34,35,36,37], or higher dysfunction severity, or even a distinctive pattern of perceptual–motor difficulties [38,39]. The diversity of the protocol used probably explains this discrepancy as the performances of the children might be highly task-dependent. For instance, Blais et al. [40] tested the SRTT protocol using letters, visuo-spatial cues or combined letters and visuo-spatial cues with children with DCD, RD and DCD-RD. Their results show that learning was never completed with letters. For children with RD, it was completed with visuo-spatial stimuli alone or added to the letter, whereas for children with DCD and children with DCD-RD, learning was only completed with combined visuo-spatial and letter stimuli. Concerning motor adaptation, we only found one study examining the effects of comorbidity between DCD and RD. Brookes et al. [41] used a prism adaptation task with children with DCD, RD, or associated DCD and RD. The participants had to throw clay balls on a large target board, wearing prismatic glasses that deflected vision laterally for 16.7°. All children adapted to the prism but with a lower rate in DCD, RD, and DCD-RD. As compared to TD children, all 8 DCD children, 5 of the 6 children with DCD-RD, and 10 of the 14 RD children showed an impaired rate of adaptation. The aftereffect was similar for all children. The authors concluded on a common deficit of the cerebellum functioning in DCD and RD, supporting an important overlap between developmental disorders. It should be noted that this visuomotor adaptation task, just as the other ones, relies on an experimental incongruence between visual and somatosensory information (e.g., prism adaptation or drawing tasks with a mirror). This may give rise to some limitations concerning the generalization of results, which may arise from a multisensory integration or sensory weighting deficit rather than from a procedural learning deficit.

This study aimed to document the link between procedural learning and neurodevelopmental disorder and compare the performance of children with isolated or associated RD and DCD in a graphomotor adaptation task. Comparing children with isolated or comorbid RD and DCD on a common graphomotor task could help to jointly address the procedural learning deficit hypothesis as a common impairment of neurodevelopmental disorders [5], and the distinction between linguistic and perceptual–motor impairments explaining graphomotor disorders in DCD and RD [42]. We compared children’s performance when writing three letters in the conventional (left to right) or opposite (right to left) writing direction. We hypothesized that the fine analysis of the movement process and product could lead to delineating the effect of the adaptation task in children with and without developmental disorders, but also to distinguish its effect between DCD and RD children. As these neurodevelopmental disorders often appear in a comorbid manner, we also analyzed the performances of children with DCD-RD compared to the performances of the former groups.

## 2. Materials and Methods

### 2.1. Participants

In total, 23 children with RD, 16 children with DCD, 19 children with RD and DCD, and 21 children with TD were included in the study (DYSTAC-MAP Cohort, Aix-Marseille). Children with neurodevelopmental disorders were recruited by the medium of public announcement or via the speech or psychomotor therapist with whom they were undergoing rehabilitation, or at the learning disability reference center in Marseille. All children received their diagnosis from a medical multi-professional team, including a neuropediatrician. TD children were recruited by means of public announcement. The parents and children gave their written informed consent to participate in the study before the start of the project, which was approved by the French Ethics Committee Review Board (CPP, agreement 2014-A01960-47). All children underwent psychological screening similar to that described by Jolly et al. [37]. The screening was conducted by a psychologist and a psychomotor therapist, and included intellectual abilities (WISC-IV) [43], oral language skills (EVAC [44] and ECOSSE [45]), phonological and reading skills (Alouette test [46] and ODEDYS battery [47]), and motor skills (MABC-1 [48]). The MABC-1 test consists of eight items grouped in three sections (manual dexterity, ball skills and balance). The items depended on the age of the children. They included manipulating pegs, cutting or threading, drawing, catching a ball, throwing a bean bag or ball, balancing, jumping, and walking (at different levels of difficulty according to age). None of the children presented any sign of cognitive impairment or neurological conditions that could affect their motor or reading abilities. All children had normal or corrected-to-normal vision as reported by their parents. We referred to the Full-Scale IQ (scores equal or above 70) when available and less than one year old, or to the Similarities and Pictures Concepts subtests (scaled score equal or above 7). These latter subtests belong to the Verbal Comprehension Index and to the Fluid Reasoning Index, and can be used to prevent the inclusion of children with low IQ [49,50]. Children presenting signs of SLI were not included. Finally, children with suspicion of attention deficit hyperactivity disorder (ADHD) as assessed through parent and clinician ratings on the DSM5 diagnostic criteria [6] were not included. All the children were right-handed.

Children were placed into one of the four groups (DCD, RD, DCD-RD and TD) according to their clinical history and to their motor and reading scores. Children with DCD were receiving intervention for a motor coordination problem that interfered with their daily living activities. They scored below the 15th percentile at the MABC-1 at our screening. They can therefore be considered as having a moderate DCD. Children with isolated or comorbid RD were treated for a reading problem by a pediatric speech therapist and scored significantly below the norm when reading isolated irregular words or logatoms (−1.5 SD; ODEDYS), and/or when reading a meaningless text (−1 SD; Alouette test). To avoid any overlap between groups, additional inclusion criteria included an MABC percentile score above the 20th percentile for RD and TD children and a reading score at the Alouette test above −0.5 SD for DCD and TD children. Children with DCD-RD fulfilled the criteria for both DCD and RD. Demographic and clinical criteria for the four groups (TD, RD, DCD, DCD-RD) are available in Table 1.

There was no difference in mean age or in gender proportion between the groups. It is worth noting that there was a significant difference in the Similarities subtest score of the WISC between the children of the TD group and children on the two groups with RD. This difference is not surprising due to the nature of the Similarities task, which relies strongly on lexicon knowledge [43].

### 2.2. Procedure

The experiment began with a presentation of the pen-display tablet and children were invited to test it while writing their first name. Familiarization trials were proposed where the children had to reproduce, in a rectangle (40 mm × 25 mm), a series of loops presented at the top of the screen. During the trigram task, the templates and a rectangle (40 mm × 25 mm) appeared on the screen until the child finished copying it (the experimentor pressed the space button to move to the next trial). The children were asked to reproduce the model in the rectangle, in the direction indicated by an arrow horizontally placed under the model. The arrow was either green and pointing to the right, or red and pointing to the left. In order to vary the task, the templates were trigrams “eue”, “eeu” and “uee” in cursive handwriting (see Figure 1A). Eight repetitions in two conditions—from the left to the right (conventional condition) and from the right to the left (adaptation condition) were tested. There was no time limit. Direction conditions and trigram orders were counterbalanced between children in each group.

### 2.3. Material

The trigram task was performed on a pen-display tablet (Wacom^®^ Cintiq 22HD, Wacom Europe GmbH, Düsseldorf, Germany) connected to a laptop piloted by a MatLab^®^ program (MathWorks, Natick, MA, USA) including the Psychophysics toolbox (Psychtoolbox-3 [51]). The children wrote directly on the surface of the tablet using a stylus. The software recorded the timing, position, and state of the pen tip on the tablet screen in real time at a sampling frequency of 60 Hz, and managed the display of the written traces, instructions, and writing areas on the screen.

### 2.4. Data Analysis

#### 2.4.1. Data Processing

Data processing was driven using Matlab^®^ routines. First of all, an offline visual inspection of the dynamic execution of each trial by two of the authors resulted in the identification of failed trials. The agreement between the two coders was obtained for each trial. Then, two spatial and two kinematical variables, which respectively concerned the product and process of handwriting, were extracted for the following analysis, conducted on successful trials only.

The product analysis relied on the written trace. We calculated an index of linearity and of the intra-trigram size variability (Figure 1B). The linearity index described the alignment of the bottoms of the letter in relation to a virtual line. It corresponds to the Euclidian distance between the three local minima (red stars in the Figure 1) on the y-axis and the regression line determined from these three points. The higher the index, the lower the linearity. The trigram size variability was calculated from the variability of the amplitude of each part of the W (dotted lines in the Figure 1) that constitutes the common “skeleton” of the trigrams.

The process analysis relied on two classical kinematical variables: the mean velocity (mm/s) of the trajectory traveled by the pen while it was in contact with the writing surface, between the starting point and the endpoint; and the amount of abnormal velocity peaks during the trace computed with the SNvpd method [52], as an index of movement disfluency. SNvpd corresponds to the difference between the velocity peaks counted after filtering the velocity profile with a cutoff frequency of 10 Hz and those counted after filtering the velocity profile with a cutoff frequency of 5 Hz. The higher the number of peaks, the less fluent the movement.

#### 2.4.2. Statistical Analysis

Statistical analyses were performed using JAMOVI^®^ software [53]. All significance thresholds were set to *p* < 0.05. Bonferroni’s correction was applied in case of multiple comparisons.

As the categorical dependent variable, the errors were submitted to the Generalized Linear Mixed Model (GLMM) with Condition (conventional, adaptation) and Group (TD, DCD, RD, DCD-RD) as factors, Trial (1 to 8) as the covariable and Subject and Trigram as the random effects.

The data collected with the pen-display tablet on the handwriting product and process mostly showed highly skewed distribution, and therefore, we identified outliers for each variable, in both condition and for each group, using the interquartile range method with a factor 3 of the IQR above the 75th percentile and below the 25th percentile. We identified extreme outlier trials, which were then removed. Each variable was then log-transformed. The new variables showing close to normal distribution (skewness between −1.3 and 1.3) were submitted to the Linear Mixed Model (LMM) with Condition (conventional, adaptation) and Group (TD, DCD, RD, DCD-RD) as factors, Trial (1 to 8) as the covariable and Subject and Trigram as the random effects. When the log transformation did not help to reduce the skewness, the trials were averaged in each condition (3 trigram and 8 trials) for each subject and non-parametric analysis was used on the mean value obtained for each condition on the complete set of data: Kruskall–Wallis and Dwass–Steel–Critchlow–Fligner pairwise comparison were used to test the effect of the Group and Wilcoxon test was used for the difference between conditions, first on the whole group, and then for each group independently using Bonferroni correction for multiple comparison.

## 3. Results

Mean performance for each group and each dependent variable is reported in Table 2.

### 3.1. Trials Failed

The percentage of failed trials differed between the groups (χ^2^ = 12.1, dll = 3, *p* < 0.01) and the conditions (χ^2^ = 74.4, dll = 1, *p* < 0.001,) and the interaction between them was also significant (χ^2^ = 36.6, dll = 3, *p* < 0.001, Figure 2). The post-hoc comparison showed that the difference between the conditions was significant in RD children (*p* < 0.005), DCD (*p* < 0.001) and DCD-RD children (*p* < 0.005), and not in TD children (*p* = 0.99). The difference between groups was only significant in the adaptation condition between TD and DCD (*p* < 0.05) or DCD-RD (*p* < 0.001), and between RD and DCD-RD (*p* < 0.01).

### 3.2. Product Analysis

Product analysis was computed on two variables: the variability of the letter’s amplitude and the linearity index of the trigrams. The performance of each group of children on the trigrams produced is illustrated in Figure 3.

The analysis revealed that the trigram variability index differed between the groups (F(3, 75) = 4.55, *p* < 0.01) and the conditions (F(1, 3213) = 26.62, *p* < 0.001, see Figure 3A). DCD-RD children significantly differed from TD children in the Adaptation condition (*p* < 0.05). The difference between conditions was significant for the DCD-RD children only (*p* < 0.001).

The linearity of the trigram differed between the groups (F(3, 75) = 4.6, *p* < 0.005) and the conditions (F(1, 3308) = 29, *p* < 0.001, see Figure 3B). The difference between TD and DCD or DCD-RD groups was significant during the Adaptation condition (*p* < 0.05). The difference between conditions was significant for the RD group (*p* < 0.01).

### 3.3. Process Analysis

The analysis of the writing process relied on the comparison of Velocity and SNvpd. Performance is presented in Figure 4.

The movement velocity depended on the condition, and was slower under the Adaptation direction than under the Conventional direction (22.3 ± 8.7 vs. 29.5 ± 13.4 cm/s, F(1, 3330.6) = 726.2, *p* < 0.001). The interaction between Group and Condition was significant (F(3, 3329.6) = 7.89, *p* < 0.001). TD children showed decreased velocity of 5.1 (19.2%), RD children of 7.1 (23.7%), DCD children of 8.3 (27.5%) and DCD-RD of 8.8 mm/s (27.8%). The post-hoc comparison did not show any significant difference between groups in the Conventional nor in the Adaptation condition (Figure 4A).

The movement disfluency was assessed with the SNvpd. The results show a difference according to the condition: the movement disfluency was higher under the Adaptation condition as compared to the Conventional condition (F(1, 3316.7) = 1152.3, *p* < 0.001, Figure 4B). This increase differed according to the group (F(3, 3315.8) = 10.2, *p* < 0.001). Calculated on the raw values, the amount of abnormal velocity peaks showed a 46.6% rise in TD children; this increase reached 69% and 68.6% in RD and DCD children, respectively, and 84.9% in DCD-RD children. The post-hoc comparison did not show any significant difference between groups in the Conventional nor in the Adaptation condition.

## 4. Discussion

The exploration of commonalities and differences between children presenting DCD and/or RD when performing an identical task contributes to the understanding of their pathogenesis. In this study, we compared the children’s performance when writing three letters in the conventional (from left to right) or opposite (from right to left) writing directions. Writing the trigram in the conventional direction is a relatively easy condition, and none of the variables in the present study identified any significant difference between the groups. However, trend differences seemed to emerge in the writing process: the groups with neurodevelopmental disorders tended to write faster than the TD group, but tended to present less spatial accuracy, producing trigrams in a more variable and less linear fashion. These findings had already been observed in writing tasks with DCD children, but not with RD children [37]. It is possible that the nature of the task in our study, which minimizes the influence of linguistic factors, explains the fact that children with RD were no slower than TD children.

Writing a trigram in the adaptation condition had an impact in all groups of children, but with a more important effect on children with neurodevelopmental disorders as compared to TD children. Both the product and the process of writing were altered when children were asked to write from right to left. The task acted like a new learning phase and entailed less accurate, slower, and less fluent handwriting. Not only were the children brought into a pseudo-learning phase, but they tended to write letters in the conventional direction, resulting in a number of failed trials. Three key findings emerged from the analysis of failed trials, which is the most representative of the difficulty of the adaptation task.

Firstly, the number of failed trials differed between the writing directions in the groups of children with neurodevelopmental disorders only. No effect was observed for the TD group. These failed trials can be discussed in the light of the “inhibitory deficit” hypothesis, which has already been supported in both DCD [14,15,54,55] and RD children (for a recent meta-review, see [56]). The pre-existing motor pattern to be inhibited in the adaption condition corresponds to the production of the letters in the conventional writing direction. Indeed, learning to write results in an association between the correct spatial form of a letter and the motor information related to the correct way of writing it. One of the key brain regions involved in such association between the spatial form and the motor information of letters has been identified in adults with functional neuroimaging [57]. They observed that the visual presentation of static letters that the participants know how to write activates a premotor area involved in writing, known as the graphemic/motor area [58]. More precisely, according to the conceptual framework of predictive coding [59], accurate motor control is thought to rely on efficient probabilistic inference integrating incoming sensory information with prior contextual knowledge [60]. Consequently, we hypothesize that the child had to inhibit the prior contextual knowledge related to motor information associated to the visual form of the letters to be reproduced, and to program another motor command corresponding to the same shape but produced in the opposite direction. This inhibition process, when unsuccessful, can be observed through the child’s writing dynamic, which appeared awkward and contained incoercible deviations (an illustration of a correct trial and of an incorrect trial are available in Appendix A). Even if the child succeeded in inhibiting the incorrect motor pattern, the handwriting movement remained slower and less fluent than in the conventional direction, and often presented a spatial deformation of the trigram (see SM1 for an illustration).

Secondly, children with DCD (isolated or comorbid with RD) more often failed to produce trigrams correctly in the adaptation condition compared to children without DCD (TD or isolated RD). We conclude that children with DCD have greater motor adaptation difficulties than RD children. This finding is not in line with what was expected. Indeed, the neurophysiological model by Doyon and colleagues [1,2] dissociates the involvement of the cortico-cerebellar network in the motor adaptation task and the involvement of the cortico-striatal network in the motor sequence learning task when consolidation has occurred during slow learning, as in the case of learning to write in children with an average age of over 9 years (for a review, see [61]). On the basis of this model, Nicolson and Fawcett [4] proposed a neural system topography for learning difficulties, and assumed that RD would be related to an impairment of the cortico-cerebellar network, whereas DCD would be related to an impairment of the cortico-cerebellar network. If this is the case, then children with RD should have more difficulty than children with DCD in a motor adaptation task. Contrary to this hypothesis, our results reveal that children with RD had less difficulty adapting their movement to the opposite writing direction than children with DCD. This finding affirms the implication of the cerebellum in DCD, which has been described extensively already [23,62,63,64,65,66,67].

Finally, the aim of this study was also to test if the comorbidity had a compounding impact and increased the children’s difficulties, or if it constitutes an alternate form of either RD or DCD. In this latter case, we expected to observe similarities between children with DCD-RD and either children with isolated RD or children with isolated DCD. Our results show that children with DCD-RD were the most impacted by the adaptation condition, as this group had the most failed trials in the adaptation condition. The performance of the DCD group was between that of the RD and DCD-RD groups, suggesting a cumulative effect of the two neurodevelopmental disorders in motor adaptation. In sum, children with comorbid DCD and RD resembled children with DCD in the motor adaptation task, but were slightly more impaired, confirming that the range of deficits causing the comorbidity could also increase the severity of symptoms [38,68]. This finding confirms that, depending on the tasks and variables analyzed, the effects of reading and motor skills levels may either be independent or additive [69]. This also echoes what has been reported and discussed for handwriting [37] and anticipatory postural adjustment [39]. In line with the “inhibitory deficit” hypothesis in the Bayesian perspective of motor control mentioned above, the computation of the intended action outcome and the updating of the priors in the case of prediction error seem to rely upon the prefrontal cortex, considered as a key region for this probabilistic inference, but also for executive functions [70]. This suggests that the presence of comorbidity in DCD and RD would not only affect the neural networks of the motor system, but would most likely have consequences on a wider range of neural networks that impact on procedural learning [62,65,71].

The main limitation of our study concerns the sample size, which probably prevents the observation of finer differences in the writing process and product. Moreover, we consider our writing task to be a motor adaptation task, but it is important to keep in mind that it also includes a sequential component, since writing trigrams requires strokes to be chained together. The sequential difficulty was minimized since there is no significant change between the strokes that make up the trigram, in terms of amplitude or direction of rotation, but this sequential component remains present. Furthermore, the writing task was performed on a screen. It is now established that the reduced coefficient of friction between the stylus and the screen disrupts writing in children [72]. Performing this task on paper would perhaps enable children to more closely feel their movement and rely more on this proprioceptive feedback to perform the adaptation task more successfully. Finally, one could consider the difference between the groups of children with single or associated RD and the group of TD children in the IQ similarity subtest as a drawback of the study. This difference is without doubt due to the fact that RD has an impact on lexico-semantic abilities [73], and therefore on the children’s performance in this subtest [43]. Nonetheless, this difference cannot be interpreted as a difference in intelligence because intelligence per se is never calculated on one WISC subtest, but on the five indexes of the scale (Verbal Comprehension, Visual Spatial, Fluid Reasoning, Working Memory, and Processing Speed).

## 5. Conclusions

This study contributes to the understanding of motor control deficits in the context of DCD and/or RD. On the one hand, the results reveal that children with RD presented a motor adaptation deficit, aligning with the cerebellar deficit hypothesis [74]. On the other hand, contrary to the hypotheses of Nicolson and Fawcett [4,5], who suggest that motor adaptation would be affected in RD, and not in DCD, our results reveal that children with RD have fewer difficulties adapting their movements than children with DCD. Finally, the greater number of failed trials in children with comorbid DCD and RD suggests a cumulative effect of the two neurodevelopmental disorders in motor adaptation. Carrying out this study using functional neuroimaging would enable us to better delineate the respective contribution of each neural network and the respective impact of each neurodevelopmental disorder on its motor system networks.

## Figures and Tables

**Figure 1 children-11-00491-f001:**
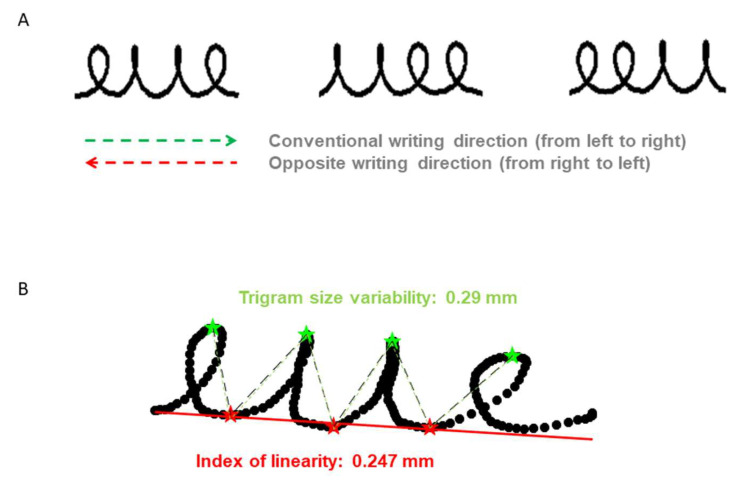
In (**A**), presentation of the three trigrams to trace in the conventional or opposite writing direction and in (**B**), illustration of the spatial variables for the product analysis of the trace.

**Figure 2 children-11-00491-f002:**
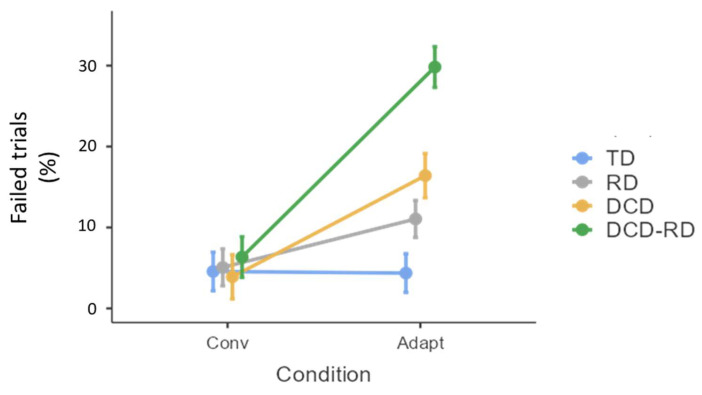
Mean percentage of failed trials according to the group. Error bars represent standard error. TD, typically developing group; RD, group with reading disorder; DCD, group with developmental coordination disorder; DCD-RD, group with associated reading disorder and developmental coordination disorder; Conv, conventional condition; Adapt, Adaptation condition.

**Figure 3 children-11-00491-f003:**
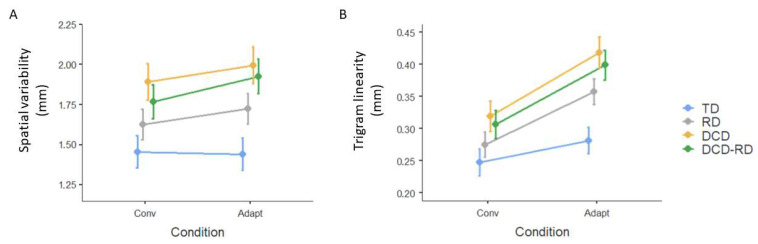
Mean value of the trigram variability index (**A**) and linearity (**B**) before Log transformation according to the group and condition. Error bars represent standard error. TD, typically developing group; RD, group with reading disorder; DCD, group with developmental coordination disorder; DCD-RD, group with associated reading disorder and developmental coordination disorder; Conv, conventional condition; Adapt, Adaptation condition.

**Figure 4 children-11-00491-f004:**
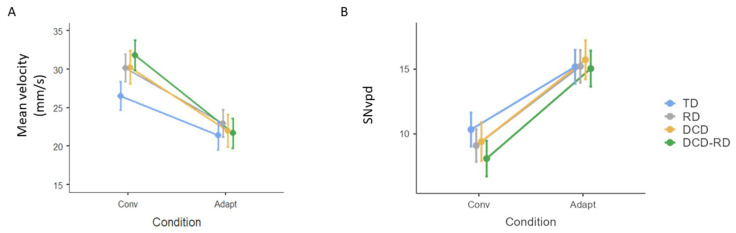
Mean velocity (**A**) and movement disfluency (SNvpd) (**B**) before Log transformation according to the group and condition. Error bars represent standard error. TD, typically developing group; RD, group with reading disorder; DCD, group with developmental coordination disorder; DCD-RD, group with associated reading disorder and developmental coordination disorder; Conv, conventional condition; Adapt, Adaptation condition.

**Table 1 children-11-00491-t001:** Participants’ characteristics.

Group		TD	DCD	RD	DCD-RD	*p*	
Number, Gender and Age	N (female)	21 (10)	16 (6)	23 (9)	19 (7)	ns	
	Age—Mean (SD)	10 (0.9)	10 (1.3)	10.1 (1.2)	9.8 (1.3)	ns	
Intelligence (WISC)	Similarities (scaled score)—Mean (SD)	16.3 (2.33)	14.1 (3.87)	13.2 (2.15)	11.7 (3.36)	ns	
	Picture Concepts (scaled score)—Mean (SD)	10.76 (1.95)	10.13 (2.55)	11.65 (2.39)	10.21 (2.27)	<0.001	DCD-RD; RD < TD
Motor (MABC-1)	Total Mean perc (SD)	46.7 (25)	4.1 (3.3)	34.8 (19.2)	3.5 (2.9)	<0.001	DCD-RD; DCD < TD; RD
	Dexterity (% chidren < 15th perc)	9.5	75	30.4	68.4	<0.005	
	Ball skills (% children < 15th perc)	14.3	50	8.7	47.4	<0.001	
	Balance (% children < 15th perc)	0	100	0	100	<0.001	
Reading	Alouette z score—Mean (SD)	1.20 (0.82)	0.40 (0.73)	−1.50 (0.72)	−1.55 0.50)	<0.001	DCD-RD; RD < TD; DCD
	ODEDYS non-words—Mean (SD)	0.72 (0.59)	0.16 (0.64)	−1.67 (1.32)	−1.47 (1.18)	<0.001	DCD-RD; RD < TD; DCD
	ODEDYS irregular words—Mean (SD)	0.97 (0.59)	0.66 (0.82)	−1.50 (0.97)	−1.69 (1.07)	<0.001	DCD-RD; RD < TD; DCD

Note: perc = percentile. Comparisons between groups were conducted using ANOVA’s and Fisher’s exact test.

**Table 2 children-11-00491-t002:** Descriptive statistics of the dependent variable for each group.

Dependant Variable	Condition	TD (Mean, SD)	RD (Mean, SD)	DCD (Mean, SD)	DCD-RD (Mean, SD)
Trials failed	Conv	4.56 (20.9)	5.07 (22)	3.91 (19.4)	6.37 (24.5)
	Adapt	4.37 (20.5)	11.1 (31.4)	16.4 (37.1)	29.8 (45.8)
Spatial variability (mm)	Conv	1.45 (0.86)	1.62 (0.84)	1.89 (1.02)	1.75 (1.02)
	Adapt	1.44 (0.66)	1.73 (0.81)	1.95 (0.99)	1.87 (0.79)
Trigram linearity (mm)	Conv	0.25 (0.2)	0.27 (0.22)	0.32 (0.26)	0.31 (0.25)
	Adapt	0.28 (0.23)	0.36 (0.29)	0.41 (0.36)	0.39 (0.33)
Mean velocity (mm/s)	Conv	26.5 (10.2)	30 (13.1)	30.2 (14.3)	31.7 (15.4)
	Adapt	21.4 (7.65)	22.9 (8.71)	21.9 (8.34)	22.9 (10.2)
SNvpd	Conv	10.3 (7.14)	9.05 (8.2)	9.43 (7.72)	7.84 (6.43)
	Adapt	15.1 (7.89)	15.3 (9.89)	15.9 (7.97)	14.5 (8.9)

Note: SD, standard deviation; TD, typically developing group; RD, group with reading disorder; DCD, group with developmental coordination disorder; DCD-RD, group with associated reading disorder and developmental coordination disorder.

## Data Availability

The data presented in this study are available on request from the corresponding author due to confidentiality and privacy disclosure of the participants’ identities and authorized sharing of the data.

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
