# Peer review of "Motor Adaptation Deficits in Children with Developmental Coordination Disorder and/or Reading Disorder"

_children, 2024, doi:10.3390/children11040491_

Round 1

Reviewer 1 Report

Comments and Suggestions for Authors

Dear Authors,

Your study devotes to the up-to-date  problem. But some points need to be edited.

1.      According to your manuscript, the research focuses on the study of the state of graphomotor skills in children with different forms of developmental disorders. The assessment tests employed in the study  evidence of the proficiency in the previously formed writing skills. The methodology did not include training sessions. Therefore, the obtained results do not allow us to state unambiguously about the state of procedural learning. Therefore, it is recommended to correct the title of the manuscript. It does not fully correspond to the content of the manuscript.

2.      Line 174: The Methods section mentions the use of the full version of WISC, but no corresponding data is given in the results.

3.      The MABC-1 methodology needs to be described in detail.

4.      It would be better to describe in detail the counting and measurement in the SNvpd method .

5.      In line 186 there is typos: meaningless test instead of meaningless text

6.      In line 197 you wrote “This difference is not surprising due to the nature of the Similarities task which relies 196 strongly on lexicon knowledge”. But the lexicon was not assessed in your study. There is no reason to treat the difference in the Similarity subtest as related to lexicon. If this subtest was used, according to the authors, as an indicator of intelligence, it means that there were between group differences in intelligence.

7.      Subjects with DCD score below 15 percentile are included in the DCD group. But according to the MABC-1 criteria this is relevant only to borderline motor impairment but not to definite motor impairment.

8.      A detailed characterization of subjects with DCD should be given in the motor domain.

9.      In the results the description statistics on all variables should be presented.

Author Response

##### Reviewer 1

Comments and Suggestions for Authors

Dear Authors,

Your study devotes to the up-to-date problem. But some points need to be edited.

  1. According to your manuscript, the research focuses on the study of the state of graphomotor skills in children with different forms of developmental disorders. The assessment tests employed in the study evidence of the proficiency in the previously formed writing skills. The methodology did not include training sessions. Therefore, the obtained results do not allow us to state unambiguously about the state of procedural learning. Therefore, it is recommended to correct the title of the manuscript. It does not fully correspond to the content of the manuscript.

Response: We would like to thank Reviewer1 for considering our manuscript positively.

The Reviewer is right, the title has been replaced by:

“Motor adaptation deficits in children with Developmental Coordination Disorder and/or Reading disorder”

  1. Line 174: The Methods section mentions the use of the full version of WISC, but no corresponding data is given in the results.

Response: The full version of the WISC was not been carried out by us. When the full IQ was available in the child’s clinical file, we did not propose the similarities and picture concepts subtests. When this was not the case, the child’s score at the similarities and picture concepts subtests was collected and used for the results reported in Table 1. This clarification has been added in the text (line 172): “We referred to the Full-Scale IQ (scores equal or above 70) when available, or to the Similarities and Pictures Concepts subtests (scaled score equal or above 7).”

  1. The MABC-1 methodology needs to be described in detail.

Response: More information about the MABC-1 methodology has been added in the Material and Methods section (line 165):

“The MABC-1 test consists of eight items grouped in three sections (manual dexterity, ball skills and balance). The items depended on the age of the children. They included manipulating pegs, cutting or threading, drawing, catching a ball, throwing a bean bag or ball, balancing, jumping, and walking (at different levels of difficulty according to age).”

  1. It would be better to describe in detail the counting and measurement in the SNvpd method .

Response: More information about the SNvpd has been added in Data processing section (line 244):

“SNvpd corresponds to the difference between the velocity peaks counted after filtering the velocity profile with a cutoff frequency of 10 Hz and those counted after filtering the velocity profile with a cutoff frequency of 5 Hz. The higher the number of peaks, the less fluent the movement.”

  1. In line 186 there is typos: meaningless test instead of meaningless text

Response: This typo has been corrected.

  1. In line 197 you wrote “This difference is not surprising due to the nature of the Similarities task which relies 196 strongly on lexicon knowledge”. But the lexicon was not assessed in your study. There is no reason to treat the difference in the Similarity subtest as related to lexicon. If this subtest was used, according to the authors, as an indicator of intelligence, it means that there were between group differences in intelligence.

Response: Even though there is a significant difference between the groups in the Similarity subtest, we would not interpret it as a difference in intelligence because intelligence per se is never calculated on one WISC subtest. This tool provides a full IQ and five index scores (Verbal Comprehension Index, Visual Spatial Index, Fluid Reasoning Index, Working Memory Index, and Processing Speed Index) computed from a large set of subtests. To put it another way, only a difference between full IQs can be interpreted as a difference ‘in intelligence’.

In the present study, the WISC subtests were used to exclude children whose low IQ could better explain the developmental difficulties than RD or DCD. The comparison of the scores was only driven for methodological transparency and should not be interpreted as a difference in intelligence between the groups.

We added the following point to the limit of the study (line 428):

“Finally, one could consider the difference between the groups of children with single or associated RD and the group of TD children at the IQ similarity subtest as a draw-back of the study. This difference is without doubt due to the fact that RD has an impact on lexico-semantic abilities [73], and therefore on the children’s performance on this subtest [43]. Nonetheless, this difference cannot be interpreted as a difference in intelligence because intelligence per se is never calculated on one WISC subtest but on the five indexes of the scale (Verbal Comprehension, Visual Spatial, Fluid Reasoning, Working Memory, and Processing Speed).”

  1. Cappelli, G. The Impact of Dyslexia on Lexico-Semantic Abilities: An Overview. In A linguistic approach to the study of dyslexia, Cappelli, G., Noccetti, S., Eds.; Multilingual Matters, Bristol, 2022; pp. 211-239.

  1. Subjects with DCD score below 15 percentile are included in the DCD group. But according to the MABC-1 criteria this is relevant only to borderline motor impairment but not to definite motor impairment.

Response: Contrarily to many studies, in the present work, children with a neurodevelopmental disorder were included through referral of their therapist because they had a RD or/and a DCD diagnosis. We did not use the MABC score as a diagnostic criterion but rather as a confirmation of the children’s motor difficulties using a common standardized test. From this point of view, we had decided to use the term DCD without any precision (moderate/severe, borderline/definite, at risk, probable…). However, we have taken this comment into account and added in the Participants section that “All children received their diagnosis from a multi-professional team, including a neuropediatrician” (line 157) and that “Children were placed into one of the four groups (DCD, RD, DCD-RD and TD) according to their clinical history and to their motor and reading scores. Children with DCD were receiving intervention for a motor coordination problem which interfered with their daily living activities. They scored below the 15th percentile at the MABC-1 at our screening. They can therefore be considered as having a moderate DCD.” (line 180).

  1. A detailed characterization of subjects with DCD should be given in the motor domain.

Response: We thank the Reviewer for this remark. To better characterize the participants with DCD without overloading the text, we opted for adding the sub-scores of the MABC-1 expressed in percentile. In each group, we counted the children having a score below or above the 15th percentile and provided the p-value of a Fisher exact test applied to the contingency table. The Table 1 was completed accordingly.

  1. In the results the description statistics on all variables should be presented.

Response: The description statistics have been added in a new table (Table 2) in the Results section (line 275).

Reviewer 2 Report

Comments and Suggestions for Authors

This manuscript presents an interesting study investigating motor adaptation in children with DCD and RD. They found performance to be poorer in those with DCD, and successively worse in those with DCD+RD. The authors should be commended on an interesting study that builds on available evidence regarding procedural learning in those with DCD. I have only minor comments:

Introduction

As a general note, the introduction provides a strong justification for the study, but it is too long in my opinion. I would suggest reducing the word count to improve readability.

The summary of work focusing on the SRTT could be shorter to improve readability. I would also suggest that this paragraph be broken into two separate paragraphs: one focusing on sequence learning, and the other motor adaptation.

Given that this study did not involve any neuroscientific measures, I’m not sure that it can be stated that the propose study would provide insight into the brain structures involved in procedural learning problems in neurodevelopmental disorders. Any such argument would be speculative without the accompanying brain measures.

Methods and Results

These were sound

Author Response

##### Reviewer 2

Comments and Suggestions for Authors

This manuscript presents an interesting study investigating motor adaptation in children with DCD and RD. They found performance to be poorer in those with DCD, and successively worse in those with DCD+RD. The authors should be commended on an interesting study that builds on available evidence regarding procedural learning in those with DCD. I have only minor comments:

 Introduction

As a general note, the introduction provides a strong justification for the study, but it is too long in my opinion. I would suggest reducing the word count to improve readability.

Response: The introduction has been reduced.

The summary of work focusing on the SRTT could be shorter to improve readability. I would also suggest that this paragraph be broken into two separate paragraphs: one focusing on sequence learning, and the other motor adaptation.

Response: The studies on the SRTT have been reduced and this paragraph has been split into two separate paragraphs accordingly. 

Given that this study did not involve any neuroscientific measures, I’m not sure that it can be stated that the propose study would provide insight into the brain structures involved in procedural learning problems in neurodevelopmental disorders. Any such argument would be speculative without the accompanying brain measures.

Response: We were unable to find this statement in our manuscript. We have reported this statement with regard to the studies conducted by Nicolson and colleagues, but for the sake of clarity, we have removed these words in the revised version of the manuscript.

Methods and Results

These were sound

Response: We thank the Reviewer for this positive comment.

Reviewer 3 Report

Comments and Suggestions for Authors

The present study compared performance of children with DCD, RD and DCD-RD on a graphomotor adaptation task.  The findings challenged the hypothesis of Nicolson and Fawcett and provides evidence of the combined impact of DCD & RD on motor adaptation.  The study has the obvious limitations that occur when working with clinical populations, but the study is well designed, appropriately analysed and the manuscript is generally very clearly presented.

I have some comments that I hope will support improvement of the manuscript.  During my reading, I wondered if the data had been collected many years ago and the manuscript also written a long time ago with some updates.  Many of my comments relate to this point.

From the reference list, I can see that much of the literature consulted relates specifically to (developmental) dyslexia rather than RD.  However, dyslexia does not appear to be mentioned in the manuscript.  It would be helpful to make clear when evidence from dyslexia research is used or provide more information early in the manuscript of the types of evidence to be used regarding RD.

I wondered if the literature consulted had been updated.  There didn’t seem to be many studies published within the last 5 years and many of the more recent publications are included in the Discussion rather than the Introduction.  The studies therefore are not informing the direction of the research.  Some recent publications that may be useful to include are listed below.

Jolly, Caroline, Marianne Jover, and Jérémy Danna. "Dysgraphia differs between children with developmental coordination disorder and/or reading disorder." Journal of Learning Disabilities (2023): 00222194231223528.

Nemmi, F., Cignetti, F., Vaugoyeau, M., Assaiante, C., Chaix, Y., & Péran, P. (2023). Developmental dyslexia, developmental coordination disorder and comorbidity discrimination using multimodal structural and functional neuroimaging. Cortex, 160, 43-54.

Dionne, Eliane, et al. "Academic challenges in developmental coordination disorder: a systematic review and meta-analysis." Physical & occupational therapy in pediatrics 43.1 (2023): 34-57.

Line 165 – What type of professional diagnosed the participants?

What were the qualifications of the researcher/s who conducted the testing? 

The French edition of the WISC-V was published in 2015.  I wondered why the WISC-IV was used.  I noticed that ethics approval for the project was dated 2014 and wondered if the data were collected at around that time (approx. 10 years ago).  It would be helpful to know more about this as the time delay seems unusual.  Were the data originally collected for another project?  Were the data for all groups collected at the same time?  More explanation is needed.

What is the reason for the focus on the Similarities and Picture Concepts subtests?  This doesn’t seem consistent with previous research e.g. Sumner, Emma, Michelle L. Pratt, and Elisabeth L. Hill. "Examining the cognitive profile of children with Developmental Coordination Disorder." Research in developmental disabilities 56 (2016): 10-17.

Does the comment in Line 174 mean that not all children completed the full WISC-IV (We referred to the Full-Scale IQ, when available..) – I’m also unclear on what this comment means.  Does it relate to cognitive impairment?

Please provide all WISC-IV subscale and Full Scale scores in Table 1.

Are the scores in Table 1 scaled scores? (i.e. not raw scores)

Please provide the scores or percentiles for the MABC, including Manual Dexterity, Ball Skills and Balance.

Line 172 – The term mental retardation is no longer used.  Perhaps use cognitive impairment.

Line 173 – I couldn’t see that any of the tests used could detect visual impairment.

Line 393 – I think the word ‘stated’ is a mistake in this sentence.  I couldn’t figure out the intended word.

There may be additional limitations when more information about measures/diagnostics is provided.  It would be helpful to consider these potential limitations in revisions.

Thank you for the opportunity t review this manuscript.

Author Response

##### Reviewer 3

Comments and Suggestions for Authors

The present study compared performance of children with DCD, RD and DCD-RD on a graphomotor adaptation task.  The findings challenged the hypothesis of Nicolson and Fawcett and provides evidence of the combined impact of DCD & RD on motor adaptation.  The study has the obvious limitations that occur when working with clinical populations, but the study is well designed, appropriately analysed and the manuscript is generally very clearly presented.

I have some comments that I hope will support improvement of the manuscript.  During my reading, I wondered if the data had been collected many years ago and the manuscript also written a long time ago with some updates.  Many of my comments relate to this point.

From the reference list, I can see that much of the literature consulted relates specifically to (developmental) dyslexia rather than RD.  However, dyslexia does not appear to be mentioned in the manuscript.  It would be helpful to make clear when evidence from dyslexia research is used or provide more information early in the manuscript of the types of evidence to be used regarding RD.

Response: The Reviewer 3 raises a relevant comment on a long-running debate about the terms “dyslexia”, “reading disability”, “reading disorder” that we find in the literature.

Actually, dyslexia is a reading disorder. For the sake of clarity, we have opted to the concept of “Reading Disorder” that we find consistent with the terms of the DSM, and more adapted: “Reading Disorder” is a better counterpart to the term “Developmental Coordination Disorder” and is in the vein of neurodevelopmental disorders. Finally, note that this term was also chosen to be in line with a previous study conducted with the same cohort (Jolly et al., 2024).

But the Reviewer is right, and accordingly, we provided more information early in the manuscript to justify the term RD rather than dyslexia (line 53):

“Nicolson and Fawcett [4,5] proposed that procedural learning deficit could constitute the core underlying dysfunction in neurodevelopmental disorders and explain the frequent comorbidity between Developmental Coordination Disorder (DCD) and dyslexia. Considering dyslexia as a reading disorder, and for the sake of clarity, we have opted throughout the manuscript for the term ‘Reading Disorder’ (RD) which is a better counterpart to DCD and consistent with the terms used in the DSM-5.”

I wondered if the literature consulted had been updated.  There didn’t seem to be many studies published within the last 5 years and many of the more recent publications are included in the Discussion rather than the Introduction.  The studies therefore are not informing the direction of the research.  Some recent publications that may be useful to include are listed below.

Jolly, Caroline, Marianne Jover, and Jérémy Danna. "Dysgraphia differs between children with developmental coordination disorder and/or reading disorder." Journal of Learning Disabilities (2023): 00222194231223528.

Nemmi, F., Cignetti, F., Vaugoyeau, M., Assaiante, C., Chaix, Y., & Péran, P. (2023). Developmental dyslexia, developmental coordination disorder and comorbidity discrimination using multimodal structural and functional neuroimaging. Cortex, 160, 43-54.

Dionne, Eliane, et al. "Academic challenges in developmental coordination disorder: a systematic review and meta-analysis." Physical & occupational therapy in pediatrics 43.1 (2023): 34-57.

Response: The first two references are already included in the list of references and the first reference appear in the Introduction section. The third reference has been included in the Introduction section (line 66, reference 10).

Line 165 – What type of professional diagnosed the participants?

Response: As the term professional was too broad, we replaced it with “All children received their diagnosis from a medical multi-professional team, including a neuropediatrician.” (line 157). As we clarified the recruitment procedure in the revised manuscript, it should be clearer now that the children took part in the study after they had been diagnosed.

What were the qualifications of the researcher/s who conducted the testing?

Response: The professionals who conducted the testing were a psychologist and a psychomotor therapist. This information has been added in the Participants section (line 162).

The French edition of the WISC-V was published in 2015.  I wondered why the WISC-IV was used.  I noticed that ethics approval for the project was dated 2014 and wondered if the data were collected at around that time (approx. 10 years ago).  It would be helpful to know more about this as the time delay seems unusual.  Were the data originally collected for another project?  Were the data for all groups collected at the same time?  More explanation is needed.

Response: This is true, the data had been collected between 2014 and 2016, as part of a large-scale project, called DYSTAC-MAP, involving several tasks including the one reported here. As noticed by the Reviewer, once ethical validation was obtained in 2014, it was no longer possible to modify the tests carried out for the sake of compliance. Furthermore, we reported in the Participants section that the children come from the DYSTAC-MAP cohort, Aix-Marseille. Concerning the publications resulting from analyses carried out with this cohort, it was clearly stated that the participants come from this DYSTAC-MAP cohort, for the sake of transparency (e.g., Bellocchi et al., 2021; Jolly et al., 2024).

Bellocchi, S., Ducrot, S., Tallet, J., Jucla, M., & Jover, M. Effect of comorbid developmental dyslexia on oculomotor behavior in children with developmental coordination disorder: A study with the developmental eye movement test. Human Movement Science 2021, 76, 102764. https://doi.org/10.1016/j.humov.2021.102764

Jolly, C.; Jover, M.; Danna J. Dysgraphia differs between children with developmental coordination disorder and/or reading disorder. J. Learn. Disabil. 2024, 222194231223528. doi: 10.1177/00222194231223528.

What is the reason for the focus on the Similarities and Picture Concepts subtests?  This doesn’t seem consistent with previous research e.g. Sumner, Emma, Michelle L. Pratt, and Elisabeth L. Hill. "Examining the cognitive profile of children with Developmental Coordination Disorder." Research in developmental disabilities 56 (2016): 10-17.

Response: Calculating the full IQ of a cohort of typically developing children can be considered as an ethical issue. To avoid any debate concerning this point, we decided to use the Similarities and Picture Concepts subtests scaled scores to exclude children scoring below 7 for a suspicion of low IQ. These subtests belong to the Verbal Comprehension and to the Fluid Reasoning Index respectively. The choice of these subtests was based on the following articles (added in the revised manuscript, line 176):

Grégoire, J. L’examen clinique de l’intelligence de l’enfant : Fondements et pratique du WISC IV [The clinical examination of children's intelligence: Foundations and practice of the WISC IV]. Sprimont : Mardaga, 2006.

Keith, T.Z., Fine, J.G., Taub, G.E., Reynolds, M.R., Kranzler, J.H. Higher order, multisample, confirmatory factor analysis of the Wechsler Intelligence Scale for Children—Fourth Edition: What does it measure. School Psychol. Rev. 2006, 35(1), 108-127.

We indeed avoid, as recommended by Sumner and colleagues, the Processing speed index as it is also often failed by children with neurodevelopmental disorder.

Does the comment in Line 174 mean that not all children completed the full WISC-IV (We referred to the Full-Scale IQ, when available..) – I’m also unclear on what this comment means.  Does it relate to cognitive impairment?

Response: We are very sorry for the lack of clarity of this method information. All the children completed the Similarities and Picture concepts subtests of the WISC, as we sought to exclude children with low IQ. For ethical reasons, the children did not do the complete IQ scale. However, children with neurodevelopmental disorder were clinically referred and some of them had complete the WISC for less than a year. In this case, the full IQ was available from the child clinical folder, and we only included children with a full IQ equal to or above 70. The score of the child at the Similarities and Picture concepts subtests was used for the results reported in Table 1. This precision was added into the text (line 172):

“We referred to the Full-Scale IQ (scores equal or above 70) when available and less than one year old, or the Similarities and Pictures Concepts subtests (scaled score equal or above 7).”

Please provide all WISC-IV subscale and Full Scale scores in Table 1.

Response: As explain in the previous comment, the Similarities and Picture concepts subtests of the WISC were the only subtests used to exclude the children of the study when the full IQ was not available. These subtests belong to the Verbal Comprehension and to the Fluid Reasoning Index respectively. In the French version of the WISC-IV, they have demonstrated good reliability (.77 and .64) and convergent validity (.58 and .50 respectively) and are considered complementary measures of general intelligence (Grégoire et al., 2006; Keith et al., 2006).

We added the following sentence in the manuscript (line 172):

“We referred to the Full-Scale IQ (scores equal or above 70), when available and less than one year old, or the Similarities and Pictures Concepts subtests (scaled score equal or above 7). These latter subtests belong to the Verbal Comprehension Index and to the Fluid Reasoning Index and can be used to prevent the inclusion of children with low IQ [49, 50].”

  1. Grégoire, J. L’examen clinique de l’intelligence de l’enfant : Fondements et pratique du WISC IV [The clinical examination of children's intelligence: Foundations and practice of the WISC IV]. Sprimont : Mardaga, 2006.
  2. Keith, T.Z., Fine, J.G., Taub, G.E., Reynolds, M.R., Kranzler, J.H. Higher order, multisample, confirmatory factor analysis of the Wechsler Intelligence Scale for Children—Fourth Edition: What does it measure. School Psychol. Rev. 2006, 35(1), 108-127.

Are the scores in Table 1 scaled scores? (i.e. not raw scores)

Response: Yes. For sake of comparability between the group, only scaled scores are provided in the Table 1. We however noticed that the term ‘standard score’ was not very frequent in English and changed the term ‘standard score’ by ‘scaled score’.

Please provide the scores or percentiles for the MABC, including Manual Dexterity, Ball Skills and Balance.

Response: The MABC total score expressed in percentile was already given in Table 1. A required, was also added cues of the groups’ scores in the Manual Dexterity, Ball Skills and Balance. In order to not overload the text, we mentioned the percentage of children, in each group, who had a score below the 15th percentile.

Line 172 – The term mental retardation is no longer used.  Perhaps use cognitive impairment.

Response: The term has been changed (line 169).

Line 173 – I couldn’t see that any of the tests used could detect visual impairment.

Response: We did not use any test to detect visual impairment, but the question was asked to the parents during the inclusion visit. We have therefore deleted the term ‘visual impairment’ and added the following sentence to the text: “All children had a normal or corrected-to-normal vision as reported by their parents” (line 171).

Line 393 – I think the word ‘stated’ is a mistake in this sentence.  I couldn’t figure out the intended word.

Response: The sentence was rewritten (line 402):

“The performance of the DCD group was between that of the RD and the DCD-RD groups, suggesting a cumulative effect of the two neurodevelopmental disorders in motor adaptation.”

There may be additional limitations when more information about measures/diagnostics is provided.  It would be helpful to consider these potential limitations in revisions.

Response: We have completed the paragraph on the limitations of our study with a discussion about the group differences in the Similarity test (line 428):

“Finally, one could consider the difference between the groups of children with single or associated RD and the group of TD children at the IQ similarity subtest as a draw-back of the study. This difference is without doubt due to the fact that RD has an impact on lexico-semantic abilities [73], and therefore on the children’s performance on this subtest [43]. Nonetheless, this difference cannot be interpreted as a difference in intelligence because intelligence per se is never calculated on one WISC subtest but on the five indexes of the scale (Verbal Comprehension Index, Visual Spatial Index, Fluid Reasoning Index, Working Memory Index, and Processing Speed Index).”

Thank you for the opportunity to review this manuscript.

Response: We thank the Reviewer for his/her evaluation.
